# Impact of B18R-Encoding Messenger Ribonucleic Acid Co-Delivery on Neutralizing Antibody Production in Self-Amplifying Messenger Ribonucleic Acid Vaccines

**DOI:** 10.3390/vaccines13050537

**Published:** 2025-05-18

**Authors:** Yutao Wang, Lei Li, Min Liang, Gan Liu, Yinying Lu

**Affiliations:** 1302 Clinical Medical School, Peking University, Beijing 100039, China; 2211210746@stu.pku.edu.cn; 2Senior Department of Hepatology, The Fifth Medical Center of PLA General Hospital, Beijing 100039, China; 3Center for Synthetic and Systems Biology, Department of Automation, Tsinghua University, Beijing 100084, China; lilei2022@tsinghua.org.cn; 4Beijing Syngenbio Co., Ltd., Beijing 100176, China; m.liang@syngenbio.com; 5Syngen-Bioimmune (Qing Dao) Co., Ltd., Qingdao 266000, China; 6Beijing Syngentech Co., Ltd., Beijing 100176, China; 7Department of Otolaryngology Head and Neck Surgery, Beijing Tongren Hospital, Capital Medical University, Beijing 100730, China; 8Comprehensive Liver Cancer Centre, The Fifth Medical Center of PLA General Hospital, Beijing 100039, China

**Keywords:** self-amplifying mRNA, chromatography, B18R, vaccine, lipid nanoparticles

## Abstract

**Objectives**: The COVID-19 pandemic has brought mRNA vaccines to the forefront due to their widespread use. In this study, we explored the potential advantages of the self-amplifying mRNA (saRNA) vaccine over conventional mRNA vaccines. **Methods**: Initially, we optimized lipid nanoparticle formulations and employed dT20 affinity chromatography purification to improve the intracellular expression of saRNA. Subsequently, we demonstrated that saRNA exhibited sustained expression for up to one month, both in vitro and in vivo, in contrast to mRNA. Finally, we developed a saRNA-based COVID-19 vaccine and achieved superior immune protection in mice compared to mRNA vaccine by co-delivering the B18R-encoding mRNA. **Results**: The co-delivery of B18R-mRNA with the saRNA vaccine significantly enhanced neutralizing antibody responses, outperforming those induced by the mRNA vaccine alone. This co-delivery strategy effectively regulated the early innate immune activation triggered by saRNA, facilitating a more robust adaptive immune response. **Conclusions**: The optimization strategies we used in this study highlight the potential of saRNA vaccines to offer stronger and more durable immune protection. The insights gained from this study not only promote the advancement of saRNA vaccine development but also provide practical guidance for their broader application in the fight against infectious diseases.

## 1. Introduction

The COVID-19 pandemic has highlighted the significant potential of mRNA vaccines as a flexible platform for rapid deployment in response to emerging infectious diseases [1]. Just 63 days after the release of the SARS-CoV-2 viral sequence, the first experimental mRNA vaccine was administered [2]. Self-amplifying mRNA (saRNA) vaccines, as an advanced iteration of mRNA technology, demonstrate significant advantages, particularly in achieving prolonged antigen expression through intracellular self-replication. Compared to conventional non-replicating mRNA, saRNA can encode a replicase within cells and use the saRNA as a template to synthesize more RNA copies, enabling self-amplification within cells. Research has shown that these subgenomic RNAs can accumulate to remarkably high levels in host cells, with copy numbers approaching 10^6^ [3]. This process can extend antigen expression duration, thereby enhancing the immune response.

Alphavirus genomes that have been used commonly for saRNA vaccine design include the Sindbis virus (SINV) [4], the Semliki Forest virus (SFV) [5], and the Venezuelan equine encephalitis virus (VEEV) [6]. Alphaviruses are small, enveloped, positive-sense RNA viruses that, during their natural replication cycle in the host cytoplasm, generate a large amount of subgenomic RNA encoding viral structural proteins [7]. saRNA retains this feature, incorporating not only the essential components of conventional mRNA but also nonstructural proteins (nsP1–4), which possess replicase activity, and a subgenomic promoter derived from alphaviruses to facilitate self-replication [8]. By replacing the viral structural protein sequences with antigen-encoding sequences, saRNA vaccines enable sustained antigen expression within cells. Although it is structurally distinct from conventional mRNA, saRNA is produced using a similar in vitro transcription (IVT) process [8].

However, the saRNA molecule, at approximately 8000–10,000 nucleotides, is significantly longer than non-replicating mRNA, which only has 1000–3000 nucleotides. This increased length often results in the generation of low-molecular-weight impurities due to incomplete transcription during the IVT process [9,10], leading to reduced integrity of saRNA and consequently affecting its expression efficiency. The extended length of saRNA also presents greater challenges for intracellular delivery [11,12]. Typically, the delivery efficiency of saRNA is lower than that of shorter non-replicating mRNA, further constraining its intracellular expression.

Another major challenge associated with saRNA is its higher immunogenicity [10,13], which can hinder translation efficiency. During the self-amplification process, double-stranded RNA (dsRNA) is produced and can be recognized by pattern recognition receptors such as RIG-I and MDA-5 [10]. This recognition triggers downstream signaling pathways that lead to the production and secretion of type I interferons (IFN-I) [10,13,14]. In contrast to saRNA, mRNA is non-replicating and therefore does not form dsRNA amplification intermediates. In addition, due to the short half-life and inherent instability of mRNA, the immunogenicity of saRNA is expected to be higher than that of mRNA. Additionally, there may be other specific features that contribute most significantly to its enhanced immunogenicity, such as antigen persistence, dose-sparing effects, and self-adjuvanticity [15]. The activation of the innate immune response not only suppresses saRNA translation but also accelerates its degradation, significantly impairing the expression of saRNAs and limiting their potential applications.

In this study, we elevated the intracellular delivery efficiency of saRNAs by optimizing the composition of lipid nanoparticles (LNPs). Additionally, we refined the purity of IVT-synthesized saRNA by eliminating abortive transcripts, thereby enhancing its structural integrity. As a result, our optimized LNP–saRNA formulation has demonstrated significantly improved intracellular expression, surpassing non-replicating mRNA in long-term expression. Then, we formulated LNP–mRNA-encoding B18R, a vaccinia-virus-derived IFN-I decoy receptor [16], as an immunomodulator. It has been reported that B18R significantly enhances saRNA translation efficiency by attenuating the innate immune response in immunocompetent cells [13]. Our use of saRNA encoding the receptor-binding domain (RBD) of the SARS-CoV-2 spike protein as the vaccine antigen, along with co-delivered mRNA-encoding B18R, has resulted in significantly higher neutralizing antibody titers in mouse immunization experiments compared to the conventional mRNA vaccine.

These findings emphasize the immunological advantages of co-delivering immunomodulators with saRNA vaccines, offering promising insights for future clinical applications of saRNA-based therapeutics.

## 2. Materials and Methods

### 2.1. Reagents and Enzymes

Lipid components for LNPs, including SM102, Acuitas A9 [6], JK102, DSPC, and DMG-PEG2000, were purchased from Beijing Jenkem Technology Co., Ltd. (Beijing, China). Cholesterol was obtained from AVT (Shanghai) Pharmaceutical Tech Co., Ltd. (Shanghai, China). The IVT synthesis kit was purchased from Hongene Biotech Corporation (Shanghai, China). Restriction endonucleases, polynucleotide kinase (PNK), T4 DNA ligase, and Q5 High-Fidelity DNA polymerase used in cloning experiments were purchased from New England Biolabs (Ipswich, MA, USA). Oligonucleotides were synthesized by Tsingke Co., Ltd. (Beijing, China).

### 2.2. Plasmid Construction

The saRNA backbone plasmids were constructed based on the VEEV replicon expression vector derived from the TC-83 strain. The mRNA manufacturing backbone plasmid was kindly provided by Beijing Syngentech Co., Ltd. (Beijing, China). For expression testing, firefly luciferase and enhanced green fluorescent protein (EGFP) were cloned into the saRNA backbone plasmid downstream of the subgenomic promoter using the Gibson assembly method, with a P2A linker between the two genes. The wild-type SARS-CoV-2 spike RBD and spike signal peptide sequence were derived from NC_045512.2, the tPA signal peptide sequence was synthesized based on GenBank J03520.1, and the B18R sequence was synthesized based on Merck’s plasmid product SCR728. DNA fragments were assembled and cloned into either the saRNA backbone plasmid or the mRNA manufacturing backbone plasmid using the Gibson assembly method. All plasmids were verified by Sanger sequencing to confirm the accuracy of the insert sequences and their junctions.

### 2.3. saRNA or mRNA Synthesis

During plasmid extraction, the A260/A280 ratio is maintained between 1.8 and 2.0. For linearized template preparation, agarose gel electrophoresis confirms complete enzymatic digestion through the absence of smearing. The saRNA or mRNA is generated through in vitro transcription mediated by T7 RNA polymerase from a linearized DNA template, using an IVT synthesis kit (Hongene Biotech Corporation, Shanghai, China) according to the protocol of Hongene Biotech. During the IVT production of saRNA, PCR tubes are used to precisely control the reaction temperature and ensure even heating, while the reaction time is appropriately extended to obtain more saRNA. When precipitating the IVT products, the standing time at −20 °C is properly extended, followed by multiple washes with −20 °C ethanol after centrifugation. RNA precipitation is performed under strictly controlled low-temperature conditions to mitigate thermal degradation.

### 2.4. saRNA Purification

The saRNA was purified using Monomix dT20 affinity column (Sepax Technologies, Suzhou, China). Tris, NaCl, EDTA, and DTT were dissolved in nuclease-free water to prepare the equilibration buffer (0.5 M) and wash buffer (0.1 M), adjusting the pH to 7.0 with HCl or NaOH, followed by autoclaving at 120 °C for 20 min. The chromatography system was flushed with ultrapure water, 0.5 M NaOH solution, and equilibration buffer. Next, the dT20 affinity column was connected to the chromatography system and flushed with nuclease-free water, 0.1 M NaOH solution, and equilibration buffer. Then, the saRNA sample was injected into the chromatography system with equilibration buffer, followed by washing with wash buffer, and finally this was eluted with nuclease-free water. Performing multiple injections with small sample volumes enhances purification efficiency while minimizing unnecessary material loss. The purified saRNA was concentrated to the appropriate concentration using an ultrafiltration tube with 50 kD molecular weight cutoff (Millipore, Burlington, MA, USA).

### 2.5. Lipid Nanoparticle Preparation

The LNPs were prepared using a modified procedure of a previous report [17]. Briefly, lipids were dissolved in ethanol containing ionizable lipid (SM102 or Acuitas A9 or JK102,), 1, 2-distearoyl-sn-glycero-3-phosphocholine (DSPC), cholesterol, and DMG-PEG (with molar ratios of 50:10:38.5:1.5). The lipid mixture was combined with 50 mM citrate buffer (pH4.0) containing saRNA or mRNA at a ratio of 1:3 using a microfluidic mixer (Micronano INano L, Micronanobiologics, Shanghai, China). Formulations were then diluted to 10× volume with PBS (pH7.4) and concentrated using an ultrafiltration tube with 100 kD molecular weight cutoff (Millipore). LNPs formulation involves gentle pipette mixing to prevent bubble formation. The final formulations were filtered through a 0.22 μm filter and stored at 2–8 °C until use. This was conducted while maintaining encapsulation efficiency > 90%, PDI < 0.2, negative zeta potential, and ~90% RNA integrity. Any batch exhibiting significant deviations from these specifications is immediately discarded and reproduced to guarantee LNP-RNA quality.

### 2.6. Cells

Huh7, 293FT, and Pan02 cell lines were generously provided by Beijing Syngentech Co., Ltd., and Vero cells were a kind gift from Vazyme (Nanjing, China). Huh7 is a human hepatocellular carcinoma cell line, Pan02 is a murine pancreatic carcinoma cell line, Vero cells are derived from African green monkey kidney cells, and 293FT cells are derived from human embryonic kidney cells. We selected the Huh7 cell line under guidance from the literature [18] for in vitro validation of normal expression of LNP–saRNA and LNP–mRNA. As we plan to conduct subsequent experiments in murine models, we chose the murine pancreatic carcinoma cell line Pan02 to observe saRNA and mRNA expression. The primary purpose of using Vero cells was for subsequent neutralizing antibody detection; based on previous research experience [19] combined with recommendations from Vazyme, we opted to use Vero cells for this purpose. Huh7, 293FT, and Vero cells were cultured in DMEM complete medium (Gibco, Thermo Fisher, Waltham, MA, USA) with 10% fetal bovine serum (FBS) and 1% penicillin/streptomycin (P/S), while Pan02 cells were cultured in RPMI 1640 complete medium (Thermo Fisher) with 10% FBS and 1% P/S. All cells were incubated at 37 °C in a 5% CO_2_ atmosphere and passaged once they reached >90% confluency. The transfection experiments were conducted 24 h after cell seeding.

### 2.7. Western Blot

Huh7 and Pan02 cells were transfected with 200 ng of LNP–saRNA or LNP–mRNA. After 48 h, the supernatant from the transfected Huh7 and Pan02 cells culture was collected for Western blot analysis. The protein samples were separated by SDS-PAGE using a kit (Beyotime, Shanghai, China). The proteins were probed with SARS-CoV-2 spike RBD antibody (1:1000 dilution; Sino Biological, Beijing, China), rabbit polyclonal 6X His tag^®^ antibody (1:5000 dilution; Abcam, Boston, MA, USA), and goat anti-rabbit IgG-HRP (1:2000 dilution; Abcam). The protein bands were visualized using the ChemiDoc MP Imaging System (Bio-Rad, Hercules, CA, USA).

### 2.8. RBD Concentration Detection

Huh7 and Pan02 cells were transfected with 200 ng of LNP–saRNA or LNP–mRNA encoding the SARS-CoV-2 spike RBD. After 48 h, the supernatants were collected, and RBD concentration was measured using a SARS-CoV-2 (2019-nCoV) spike RBD ELISA kit (Sino Biological) according to the manufacturer’s instructions. The detailed steps are as follows: Firstly, equilibrate all reagents to room temperature (22–28 °C) before use. Remove any unused microplate strips from the plate frame and return them to the foil pouch containing the desiccant pack and reseal. Add 300 μL of 1× wash buffer to each well and let it stand for approximately 2 min. Aspirate or dump the liquid and pat dry on a paper towel; wash twice in this way. Prepare 1000 μL of the 500 pg/mL top standard. Use 500 μL of 1× Dilution Buffer as the diluent to perform six two-fold serial dilutions of the 500 pg/mL top standard in six separate tubes. The 1× Dilution Buffer serves as the zero standard. Next, add 100 μL standard and appropriately diluted test samples per well. Seal the plate and incubate at room temperature for 2 h. After incubation, add 300 μL of 1× wash buffer to each well and let it stand for approximately 2 min. Aspirate or dump the liquid and pat dry on a paper towel; wash wells three times in this way. Add 100 μL of Detection Antibody Working Solution into each well and mix gently. Cover or seal the plate and incubate at room temperature for 1 h. Afterwards, remove the liquid from the wells and wash the plate three times. Add 100 μL of Substrate Solution (an equal mixture of Color Reagents A and B) to each well and mix gently. Incubate at room temperature for 10–20 min, protected from light. Add 100 μL of stop solution to each well and tap the plate gently to ensure it is well mixed. Read the absorbance of the entire plate at 450 nm wavelength. Calculate the data using a four-parameter logistics curve-fitting algorithm.

### 2.9. Mouse Vaccination and Serum Collection

Female BALB/c mice, aged 6–8 weeks, were randomly divided into four groups: (1) 0.5 μg saRNA vaccine group; (2) 1 μg saRNA vaccine group; (3) 1 μg mRNA vaccine group; (4) PBS placebo group. Each vaccine group included 6 mice, and the placebo group had 3 mice. The prime immunization was administered intramuscularly on day 0, followed by a boost immunization on day 14. The saRNA vaccine groups were co-administered with LNPs (B18R-mRNA). Mice were anesthetized with isoflurane at a dose of 0.41 mL/min with a fresh gas flow of 4 L/min (using a concentration of 2%). After the mice lost consciousness, blood samples were collected from the retrobulbar venous plexus using a capillary glass tube. Blood samples were collected on days 7, 14, 21, and 42 post-prime immunization, centrifuged at 3500 rpm for 1 h after standing at room temperature for 4 h, and the serum was stored at −80 °C. Institutional Animal Care and Use Committee (IACUC) of Tsinghua University, Beijing, China, approved all animal protocols used in this study (protocol code: THU-LARC-2025-004). All methods are reported in accordance with ARRIVE guidelines.

### 2.10. IgG Antibody Titer Evaluation

Serum samples were heat-inactivated at 56 °C for 30 min. IgG antibody titers were determined by ELISA assay. The plate was coated with recombinant 2019-nCoV S-trimer protein (C-6His) (Novoprotein, Suzhou, China) and incubated overnight. Serum samples from mRNA-vaccinated mice were initially diluted 1:300, followed by eight 1:6 serial dilutions. For saRNA-vaccinated mice, serum samples were initially diluted to 1:50, followed by eight 1:3 serial dilutions. After 2 h of incubation at room temperature, the plates were incubated with goat anti-mouse IgG-HRP (1:1000 dilution, BioLegend, San Diego, CA, USA) for 30 min at room temperature. After incubation with TMB substrate (BioLegend) for 5–10 min, the reaction was stopped using ELISA stop solution (Solarbio, Beijing, China), and the OD values at 450 nm were read using a microplate reader. Endpoint titers were defined according to the manufacturer’s instructions. IgG2a and IgG1 antibody titers were measured using goat anti-mouse IgG2a-HRP or IgG1-HRP by the similar ELISA method.

### 2.11. Fifty Percent Neutralizing Antibody Titer Evaluation

Serum samples were heat-inactivated at 56 °C for 30 min and diluted with DMEM medium (Gibco, Thermo Fisher). SARS-CoV-2-Fluc pseudovirus (Vazyme) was diluted to 13000 TCID_50_/_mL_ in DMEM medium. Diluted serum and pseudovirus were mixed and incubated at 37 °C with 5% CO_2_ for 1 h. Vero cells were prepared at a density of 5 × 10^5^ cells/mL. After incubation, 100 μL of the cell suspension was added to each well of vero cells, and the plates were incubated at 37 °C with 5% CO_2_ for 24 h. Next, luciferase substrate (Bright-Lumi™ II, Beyotime) was added, incubated for 3–5 min, and chemiluminescence was measured using a microplate reader. The 50% neutralization titer (NT_50_) was defined as the serum dilution at which relative light units (RLUs) were reduced by 50% compared to virus control wells. NT_50_ values were calculated by non-linear regression using GraphPad Prism 8.0.2 (GraphPad Software).

### 2.12. Statistical Analysis

All data were analyzed with GraphPad Prism 8.0.2 software. Data were presented by mean ± SD in all experiments. Analysis of variance (ANOVA) or t-test was used to determine statistical significance among different groups (* *p* < 0.05; ** *p* < 0.01; *** *p* < 0.001; **** *p* < 0.0001; ns, not significant).

## 3. Results

### 3.1. Optimization of saRNA Expression

We initiated the process by modifying the plasmid backbone for saRNA production based on the VEEV genome. This involved adapting it for IVT synthesis by integrating a sequence downstream of the T7 promoter to facilitate the binding of the cap analog. Two versions of the saRNA construct were tested, and the selection of saRNA3 was primarily based on the firefly luciferase reporter gene expression levels, with saRNA3 demonstrating the highest luminescence expression among all candidates (Appendix A). Consequently, the saRNA3 version was used for subsequent experiments, while saRNA1 and saRNA2 were not further considered in our experimental workflow. Next, we evaluated the impact of different ionizable lipids on the efficiency of intracellular delivery of saRNA using LNPs. LNPs formulated with three different ionizable lipids containing 500 ng of EGFP-encoding saRNA were added to Huh7 cells, and transfection efficiency was assessed 48 h later. The results revealed that LNPs formulated with SM102 exhibited the best delivery efficiency, achieving nearly 70% intracellular uptake (Figure 1a). Additionally, it was observed that cell growth was affected post-transfection, indicating that 500 ng per well is not the optimal dose. Consequently, we reduced the transfection dose in following experiments. The SM102-LNP displayed favorable physicochemical properties, with an average particle size of approximately 80 nm, a uniform size distribution, and an encapsulation efficiency exceeding 97% (Figure 1b). The integrity of IVT-synthesized saRNA was then assessed, which was found to be 81.8%, with 16.4% representing incomplete transcripts (Figure 1c). Following purification using dT20 affinity chromatography, the impurities were reduced to less than 5%, significantly improving saRNA integrity to 93.6% (Figure 1d). LNPs encapsulating saRNA encoding EGFP, both before and after purification, were added to Huh7 cells at a dose of 200 ng per well, and transfection efficiency (positive rate) and expression levels (mean fluorescence intensity, MFI) were assessed at 48 h and 72 h post-transfection. We established three transfection dosage groups (100 ng, 200 ng, and 300 ng) for Huh7 cells (Figure 2b). Notably, we observed no significant differences in expression levels between the 200 ng and 300 ng groups at 48 h post-transfection. Furthermore, both transfection efficiency and expression levels showed no statistical differences between these two dosage groups at 72 h post-transfection. Considering the balance between minimizing potential cytotoxic effects and achieving optimal experimental outcomes, we selected 200 ng as the optimal transfection dose. Subsequent analysis indicated a substantial enhancement in both transfection efficiency and expression levels post-purification (Figure 1e). These optimizations resulted in excellent in vitro expression capabilities for the LNP–saRNA formulation. Therefore, in all following experiments, dT20 affinity chromatography was employed for saRNA purification, and SM102 as the ionizable lipid in the LNP formulations.

### 3.2. Prolonged Expression of saRNA

Moreover, the prolonged-expression capability of saRNA was assessed. The duration of saRNA expression was assessed qualitatively within the cells. EGFP-encoding saRNA was transfected into Huh7 cells at a dose of 200 ng per well, and the fluorescence expression was continuously monitored during cell passaging. As shown in Figure 2a, saRNA expression in cells persisted for up to 32 days. To quantitatively compare the expression duration of saRNA and mRNA, the dose–response relationship of saRNA expression was evaluated first. Flow cytometry analysis revealed that both the positive rate and MFI increased in a dose-dependent manner at 48 h and 72 h post-transfection in Huh7 cells, with the positive rate exceeding 80% at 72 h (Figure 2b). Interestingly, we observed that there was a lower transfection rate of saRNA in 293FT cells (Appendix A), possibly due to different cell tolerances to saRNA, which indicates the importance of considering cell-type-specific differences in saRNA expression. Further comparison of the long-term expression of saRNA and mRNA were conducted in Huh7 cells at a dose of 200 ng LNP–saRNA and equimolar LNP–mRNA per well. Luciferase was selected as the reporter gene due to concerns that the prolonged half-life of EGFP could result in expression accumulation, potentially compromising result accuracy. Additionally, this choice ensured alignment with subsequent in vivo experiments. Luciferase expression was tracked during continuous cell passaging. Prior to passaging, cell confluence was monitored under a microscope, and passaging was initiated when confluence reached ≥90%. The resulting cell suspension was then divided equally into two portions—one aliquot was allocated for luciferase reporter gene assay, and the other was transferred to a new 24-well plate for continued passaging and culture. In a 32-day experiment, saRNA sustained high expression levels for up to one month, whereas mRNA expression markedly declined by day 11 (Figure 2c). In addition, saRNA expression was assessed in the murine Pan02 cell line prior to the in vivo tests. During cell culture, we observed significant differences in proliferation rates between Huh7 and Pan02 cells, with the latter exhibiting markedly accelerated growth kinetics. To mitigate potential detrimental effects arising from Pan02 cell overgrowth in culture wells, we strategically selected the 24 h and 48 h post-transfection timepoints for experimental assessments. Transfection with three different doses achieved positive rates exceeding 90% at 24 h, with expression levels inversely correlated with dose (Figure 2d). This suggests that saRNA exhibits excellent transfection efficiency and expression performance in murine cells, even at low doses. In subsequent in vivo experiments, mice were intramuscularly injected with 5 µg of LNP–saRNA in the left hindlimb and an equimolar amount of LNP–mRNA in the right hindlimb. This demonstrated that saRNA expression steadily increased over four weeks, while mRNA expression dropped to minimal levels within one week of intramuscular injection (Figure 2e,f). These findings confirm that saRNA offers superior long-term expression compared to mRNA, both in vitro and in vivo.

### 3.3. saRNA Vaccine Construction and Validation

To investigate the differences in immune response elicited by saRNA and mRNA vaccines, we constructed vaccines encoding the RBD of the SARS-CoV-2 spike protein, as shown in Figure 3a. The antigen secretion capacity induced by the signal peptides of the spike protein and tissue plasminogen activator (tPA) was assessed using Western blot analysis in Huh7 and Pan02 cells. The results indicated that the tPA signal peptide outperformed the spike signal peptide in both mRNA (Figure 3b) and saRNA (Figure 3c) vaccines, leading to the selection of the tPA signal peptide for further experiments. Subsequently, we measured RBD expression levels using ELISA and found that the saRNA vaccine produced significantly higher RBD levels than the mRNA vaccine in Pan02 cells (194.2 ng/mL vs. 12.14 ng/mL), though they were comparable in Huh7 cells (Figure 3d). Our preliminary experimental results indicate that the co-electroporation of saRNA and B18R-mRNA enhanced transfection efficiency (Appendix A). For the subsequent co-delivery of saRNA and B18R-mRNA, we formulated LNP-B18R-mRNA, an immunomodulator. Western blot analysis confirmed successful B18R expression in both Pan02 and Huh7 cells, with higher expression observed in Huh7 cells, likely due to codon optimization of the B18R sequence for human tissues (Figure 3e). Following the validation of expression for all RNA constructs, we proceeded with in vivo immunization studies in mice.

### 3.4. In Vivo Immunization Studies

In vivo immunization experiments were conducted using 6–8-week-old female Balb/c mice, randomly divided into four groups: (1) 0.5 μg saRNA vaccine; (2) 1 μg saRNA vaccine; (3) 1 μg mRNA vaccine; (4) PBS as a placebo. Additionally, 2 μg of B18R mRNA was co-administered with the saRNA vaccines. The experimental schedule is illustrated in Figure 4a. Four weeks after the prime-boost immunization, the titers of SARS-CoV-2 RBD-specific IgG and neutralizing antibodies, measured as the 50% neutralization titer (NT_50_), were evaluated. The IgG titers in the 0.5 μg and 1 μg saRNA vaccine groups were approximately 1/7690 and 1/1332, respectively, both significantly lower than the 1/512,679 titer observed in the 1 μg mRNA vaccine group (Figure 4b). On day 21, the IgG antibody titers in the saRNA vaccine group were lower than those in the mRNA vaccine group, mirroring the overall trend observed on day 42 (Appendix A). However, the NT_50_ in the 1 μg saRNA group reached approximately 1/30,598, substantially higher than that of the 1 μg mRNA group (1/8848) and the 0.5 μg saRNA group (1/104) (Figure 4c). This suggests that the prolonged antigen expression from the saRNA vaccine provides sustained immune stimulation, resulting in antibodies with superior neutralizing capabilities compared to those induced by the mRNA vaccine. The weaker neutralizing antibody response in the 0.5 μg saRNA group suggests that the effective dose of saRNA vaccines warrants particular attention in future applications. The immune response skewing was analyzed by calculating the IgG2a/IgG1 ratio. The 1 μg saRNA vaccine group and 1 μg mRNA vaccine group exhibited comparable IgG2a/IgG1 ratio, which peaked on day 7 post-immunization and subsequently demonstrated a gradual decline thereafter (Figure 4d). This indicates that both saRNA and mRNA vaccines elicit a similar pattern of immune response, initially skewing toward a Th1-biased response indicative of robust cell-mediated immunity, which later shifts to a balanced Th1/Th2 response as humoral immunity becomes more prominent. The in vivo immunization results suggest that, while saRNA and mRNA vaccines exhibit similar immune response profiles, the saRNA vaccine confers significantly stronger immune protection.

## 4. Discussion

In this study, we developed an optimized saRNA-based COVID-19 vaccine that leverages the persistent expression characteristics of saRNA and evaluated its advantages over mRNA vaccines. Initially, we conducted a screening process to identify the most effective LNP formulation that significantly enhances intracellular transfection efficiency of saRNA. SM102 has been incorporated into commercially approved vaccine products (e.g., Moderna’s COVID-19 vaccine) with demonstrated efficacy and safety profiles, and SM102 has also been widely utilized in numerous scientific studies [20,21,22]. Given the presence of short-fragment impurities in the IVT-synthesized long-chain saRNA bulk solution, we employed a dT20 affinity chromatography column for purification, yielding highly intact saRNA. Previous studies have reported that cellulose-based purification methods can also achieve effective saRNA purification [14,23]. Additionally, high-performance liquid chromatography (HPLC) purification can efficiently remove impurities such as dsRNA and proteins that may trigger immune responses, and HPLC-purified nucleoside-modified mRNAs demonstrated significantly improved translation efficiency [24]. Our experimental results demonstrated that these optimization strategies, including affinity chromatography, cellulose-based purification, and HPLC, all significantly enhanced the intracellular expression efficiency of LNP–saRNA. Furthermore, while modified nucleosides can substantially reduce the activation of innate immune responses in mRNA vaccines, studies have shown that such modifications appear insufficient to effectively mitigate saRNA-induced innate immune activation [25]. Therefore, further research and exploration are required to determine optimal nucleoside selection, control the ratio of modified to unmodified nucleosides, and adapt various in vitro preparation conditions for saRNA [26].

In the long-term expression validation experiments of saRNA, we qualitatively observed that transfection of saRNA into Huh7 cells via LNPs resulted in sustained expression for over one month. To quantitatively compare saRNA and mRNA, we monitored the expression of firefly luciferase during continuous cell passages. Luciferase was selected as the reporter gene instead of EGFP due to the latter’s long half-life, which could lead to intracellular accumulation and compromise the accuracy of final results in long-term experiments. Studies have shown that the average fluorescence lifetime of EGFP is influenced by factors such as pH, indicating that changes in EGFP fluorescence characteristics may correlate with intracellular microenvironments [27]. While EGFP serves as a commonly used reporter gene with advantages like intuitive detection and ease of use, its prolonged half-life can become a disadvantage in certain scenarios. When EGFP expression levels are high or prolonged, its cumulative effects may lead to the misinterpretation of experimental results. In contrast, luciferase, as an enzymatic reporter gene, directly correlates its activity with light signals generated by substrate reactions, theoretically providing more accurate reflections of gene expression dynamics. Over the 32-day experiment, saRNA demonstrated persistent high expression, whereas mRNA expression nearly vanished by day 11 post-transfection. Notably, saRNA exhibited remarkable transfection efficiency and expression performance in the murine Pan02 cell line, bolstering our confidence in testing its long-term expression in vivo. In a 28-day murine study, saRNA expression steadily increased over time, further validating its long-term expression characteristics, which were more pronounced in vivo than in vitro. Interestingly, we observed dose-dependent differences in saRNA transfection efficiency and expression levels between Huh7 and Pan02 cells, which we propose represents a particularly valuable and intriguing finding. Such disparities likely arise from the complex interplay of cell-specific elements, RNA structural determinants, and translational regulatory mechanisms that collectively modulate post-transcriptional gene expression dynamics [28,29].

To explore the potential advantages of saRNA in vaccine applications, we constructed COVID-19 vaccines featuring the SARS-CoV-2 spike protein’s RBD as the antigen, including both saRNA and mRNA versions. Prior to immunization studies in mice, we compared the secretion efficiency of the target antigen (RBD protein) promoted by tPA signal peptide versus the original spike signal peptide, revealing that tPA signal peptide significantly enhanced RBD secretion in both Huh7 and Pan02 cell lines. Previous studies have also utilized tPA signal peptide in adenovirus-based [30,31] and mRNA-based [32] COVID-19 vaccines. In subsequent murine immunization studies, we co-delivered mRNA-encoding B18R to the saRNA vaccine group. Experimental studies have shown that integrating the B18R gene into oncolytic herpes simplex virus can attenuate innate immune activation during viral replication, thereby enhancing viral stability in the tumor microenvironment and improving antitumor efficacy [33]. Alleviating innate immune activation by adding the interferon-inducible receptor B18R also significantly improved saRNA translation [13]. In an RNA-based iPSC generation study, co-transfection of VEE-reprogramming factor RNA replicon with B18R mRNA achieved high-level expression of four reprogramming factors in human fibroblasts [34]. During experimental design, we include another experimental group (1 μg saRNA without B18R-mRNA co-delivery). While the IgG antibody titers in this group showed a *p* = 0.05 statistical trend approaching significance compared to the 1 μg saRNA/B18R-mRNA co-delivery group, the co-delivery group still demonstrated a discernible advantage (Appendix A). As there are currently no published reports on the combined use of B18R-mRNA in the development of vaccines, our study represents a pioneering exploratory investigation. In preliminary experiments, concurrent administration of low-dose B18R failed to demonstrate significant enhancement of saRNA vaccine immunogenicity. However, at the elevated dosage of 2 μg, we observed a pronounced immunopotentiating effect, prompting us to standardize this dosage for all subsequent experiments. Unfortunately, the current study did not incorporate dose–response analysis to further delineate the minimum effective dose or potential toxicity profile of B18R-mRNA. This critical gap necessitates subsequent investigation to fully elucidate the therapeutic potential of B18R-mRNA as an immunomodulatory agent.

Our analysis of immune response skewing showed comparable bias and trends between saRNA and mRNA vaccine-induced immune responses. While the IgG2a/IgG1 ratio may serve as an imperfect surrogate marker, it does provide meaningful directional insights into Th1/Th2 polarization patterns. Th1 responses, primarily driven by IFN-γ secretion, orchestrate cellular immunity, whereas Th2 responses mediated by IL-4 regulate humoral immunity [35]. IgG1 production is induced by IL-4 (Th2-associated cytokine), while IgG2a synthesis requires IFN-γ (Th1-associated cytokine). Elevated IgG2a/IgG1 ratios suggest Th1 predominance, whereas reduced ratios indicate Th2 bias. Both Th1 and Th2 subsets derive from CD4*+* T cell differentiation, with Th1 cells secreting IFN-γ and TNF-β to activate macrophages and cytotoxic T lymphocytes, and Th2 cells producing IL-4/IL-5/IL-13 to promote B cell responses. CD8*+* cytotoxic T lymphocyte activity is modulated by Th1 enhancement and Th2 suppression, forming a complex immune regulatory network. Notably, multiple studies have adopted IgG2a/IgG1 ratios as relevant immunological readouts [6,36,37,38]. Following vaccine-induced immune activation, Th1/Th2 skewing exhibits dynamic changes: Th1-biased immunity prevails in early stages, while Th1/Th2 balance gradually emerges during late-phase immune induction. Notably, saRNA-based COVID-19 vaccines have been reported to induce S-specific T cell responses with Th1 bias [39].

We evaluated IgG antibody titers and NT_50_ in mice four weeks after booster immunization. While saRNA vaccines induced significantly lower IgG antibody titers than mRNA vaccines, they elicited markedly higher NT_50_ levels, indicating superior immunoprotective effects. In cell experiments, we observed that saRNA maintained high-level intracellular expression for up to 32 days, whereas mRNA expression sharply declined by day 11. In animal experiments, saRNA sustained high-level expression for four weeks, while mRNA expression dropped to very low levels after approximately one week. Based on these results, we hypothesize that prolonged antigen presentation mediated by saRNA may correlate with enhanced immune responses, leading to the generation of superior neutralizing antibodies compared to mRNA.

IgG titer refers to the concentration or level of immunoglobulin G in serum. IgG is the major component of immunoglobulins in serum, accounting for approximately 75% of the total immunoglobulin pool. NT_50_ represents the antibody concentration required to neutralize 50% of the virus, serving as a metric to describe the neutralizing capacity of antibodies against viruses, i.e., their ability to inhibit or eliminate viral infectivity. IgG antibodies comprise both neutralizing IgG antibodies and non-neutralizing IgG antibodies, meaning not all IgG antibodies possess neutralizing activity. Some reports indicate a complex correlation between IgG titers and NT_50_, rather than a strict linear relationship [40], with some reports even suggesting weak correlations [41]. We therefore hypothesize the following mechanism underlying the discrepancy between IgG titers and NT_50_: saRNA vaccines induce high-quality neutralizing antibodies through sustained antigen expression and stronger immune activation, even with lower total IgG titers; in contrast, mRNA vaccines elicit abundant low-affinity antibodies, resulting in more IgG titers in total but weaker neutralizing capacity. mRNA vaccines, characterized by transient antigen expression and dose-dependent efficacy, might fail to adequately activate B cells at suboptimal doses, leading to suboptimal affinity maturation. Follicular helper T cells (Tfh), which orchestrate B cell proliferation and class-switching through cytokine signaling (e.g., IL-21), are pivotal for antibody affinity maturation [42]. The transient antigenic stimulation from mRNA vaccines may result in insufficient Tfh cell activation, thereby limiting antibody optimization. Conversely, saRNA’s self-amplifying mechanism via RNA-dependent RNA polymerase (RdRp) enables prolonged antigen production and increased protein yields. This extended antigen exposure provides greater opportunities for B cell somatic hypermutation and selection, favoring the generation of high-affinity antibodies. Additionally, saRNA’s intrinsic adjuvant properties [43] may enhance immune cell activation, particularly Tfh-mediated B cell help, further supporting antibody affinity maturation. The specific mechanisms underlying this phenomenon warrant further investigation.

The current study also has several limitations. Firstly, regarding the dose–response analysis of B18R-mRNA co-delivery, we only tested a single dosage in this investigation. Future studies should incorporate multiple dose gradients to identify the minimum effective dose while monitoring potential adverse effects and toxicity profiles. The specific mechanisms by which B18R-mRNA regulates innate immune responses need to be investigated further. Secondly, concerning the detection of specific T cells and related cytokines, we indirectly assessed Th1/Th2 immune skewing through the IgG2a/IgG1 ratio. However, direct characterization of CD4+ T cell subsets (including Th1, Th2, Tfh cells, etc.) and CD8+ T cells, coupled with cytokine secretion analysis, would provide more definitive evidence. Thirdly, regarding IgG repertoire analysis, while our findings demonstrated enhanced neutralizing antibody production with saRNA/B18R-mRNA co-delivery, further investigation into serum antibody profiles is warranted. This includes systematic evaluation of individual IgG subclasses, detailed antibody composition analysis, and targeted studies on their protective efficacy.

## 5. Conclusions

In conclusion, this study successfully enhanced saRNA integrity through dT20 affinity chromatography purification and utilized SM102 to formulate LNPs for efficient saRNA delivery. We validated saRNA’s long-term expression capacity both in vitro and in vivo. Furthermore, in murine immunization experiments, we demonstrated that co-delivering B18R-mRNA with saRNA vaccines induced significantly more potent protective neutralizing antibodies compared to mRNA vaccines. The optimized strategies and immunization approaches described in this study provide promising insights and practical guidance for the broad application of saRNA vaccines.

## Figures and Tables

**Figure 1 vaccines-13-00537-f001:**
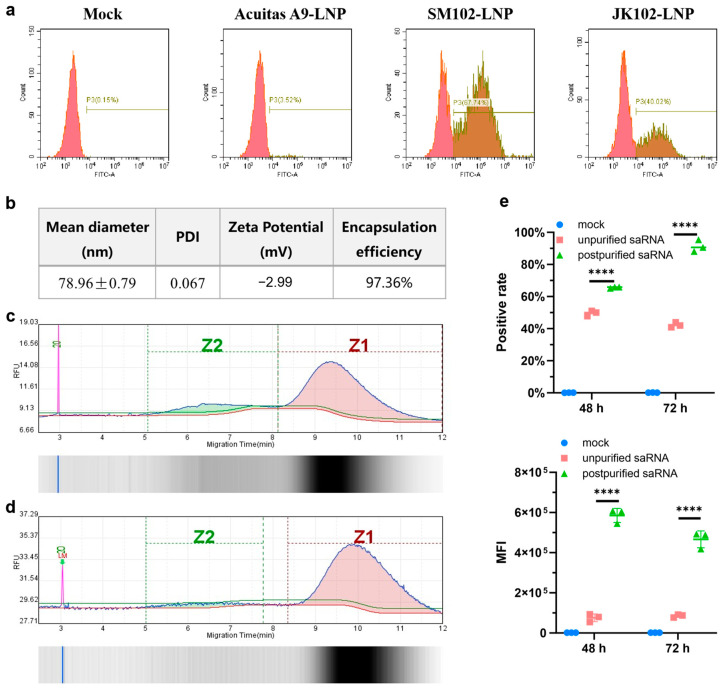
Optimization of saRNA enhances expression. (**a**) Transfection efficiency of saRNA encapsulated in LNPs formulated with different ionizable lipids in Huh7 cells. At a transfection dose of 500 ng per well, detection was performed 48 h post-transfection. (**b**) The physicochemical properties of SM102-LNP encapsulating saRNA. Integrity analysis of IVT-synthesized saRNA before purification (**c**) and after purification (**d**). Z1: Intact saRNA; Z2: Impurity from incomplete transcripts. (**e**) Comparison of positive rate and MFI in Huh7 cells using LNP–saRNA both before and after purification. At a transfection dose of 200 ng per well, detection was performed 48 h and 72 h post-transfection. **** *p* < 0.0001.

**Figure 2 vaccines-13-00537-f002:**
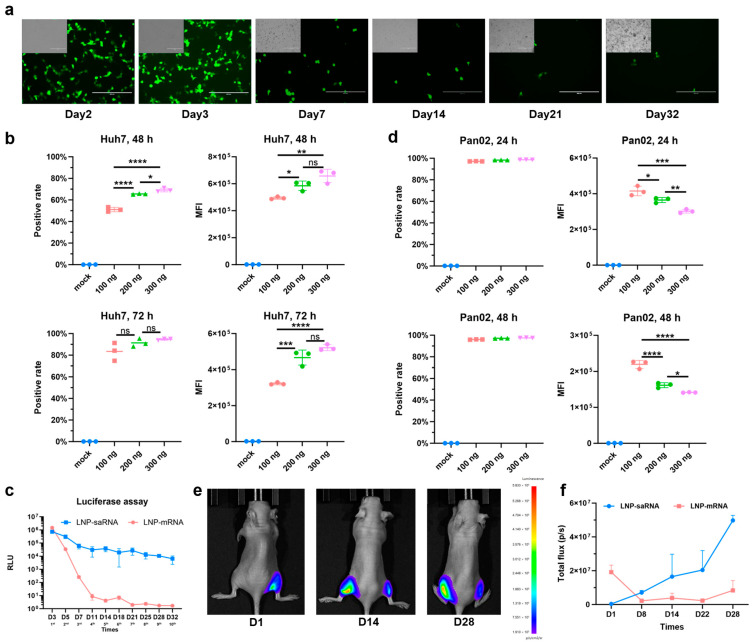
Validation of prolonged expression of saRNA. (**a**) Fluorescence of Huh7 cells transfected with saRNA at a dose of 200 ng per well was continuously monitored during cell passages. Scale bar: 400 μm. (**b**) Positive rate and MFI of varying doses of LNP–saRNA in Huh7 cells. Detection was performed 48 h and 72 h post-transfection. (**c**) Long-term expression comparison profile between 200 ng of LNP–saRNA and equimolar LNP–mRNA over multiple cell passages. Each time point corresponds to one passage, with a total of 10 consecutive passages completed. (**d**) Positive rate and MFI of varying doses of LNP–saRNA in Pan02 cells. Detection was performed 24 h and 48 h post-transfection. In vivo imaging (**e**) and quantitative comparison profile (**f**) of prolonged expression in mice, with 5 μg of LNP–saRNA injected intramuscularly into the left hindlimb muscle and an equimolar amount of LNP–mRNA injected intramuscularly into the right hindlimb. All cellular experiments were conducted using 24-well plates. * *p* < 0.05; ** *p* < 0.01; *** *p* < 0.001; **** *p* < 0.0001; ns, not significant.

**Figure 3 vaccines-13-00537-f003:**
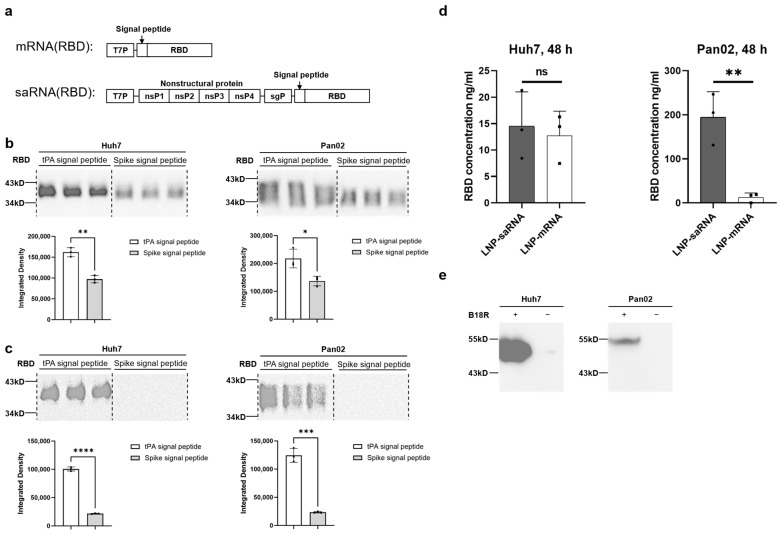
Vaccine expression analysis. (**a**) Schematic diagram of the mRNA and saRNA vaccines. Western blot analysis of different signal peptide variants of the mRNA vaccine (**b**) and saRNA vaccine (**c**) expression in Huh7 and Pan02 cells. Cell supernatants were collected 48 h after transfection with 200 ng of the vaccines. (**d**) Quantitative measurement of RBD concentrations in Huh7 and Pan02 cells transfected with the 200 ng saRNA vaccine or mRNA vaccine using the ELISA method. Cell supernatants were collected 48 h after transfection. (**e**) Western blot analysis of B18R expression in Huh7 and Pan02 cells. Cell supernatants were collected 48 h after transfection with 500 ng of the LNP–mRNA-encoding B18R. * *p* < 0.05; ** *p* < 0.01; *** *p* < 0.001; **** *p* < 0.0001; ns, not significant.

**Figure 4 vaccines-13-00537-f004:**
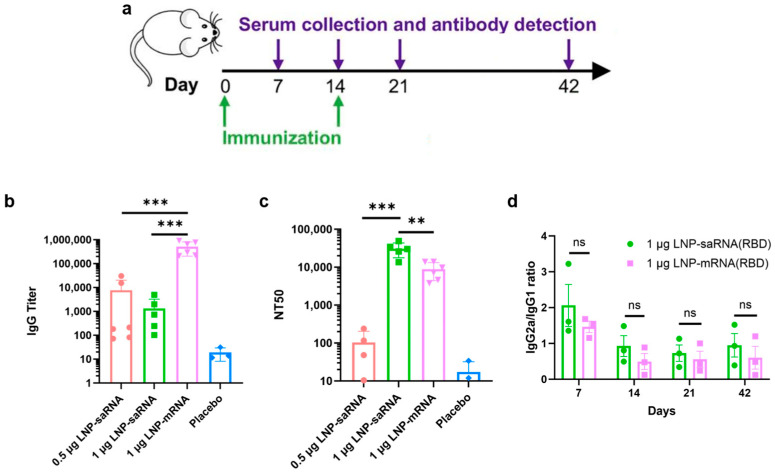
Mice immunization experiment. (**a**) Schematic diagram of the immunization schedule and sample collection. IgG antibody titer (**b**) and NT_50_ (**c**) at day 42 post prime immunization. (**d**) IgG2a/IgG1 ratio at various time points. ** *p* < 0.01; *** *p* < 0.001; ns, not significant.

## Data Availability

The raw data supporting the conclusions of this article will be made available by the corresponding authors on request.

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
