# Peer review of "Impact of B18R-Encoding Messenger Ribonucleic Acid Co-Delivery on Neutralizing Antibody Production in Self-Amplifying Messenger Ribonucleic Acid Vaccines"

_vaccines, 2025, doi:10.3390/vaccines13050537_

Round 1
Reviewer 1 Report
Comments and Suggestions for Authors
This manuscript presents the development and optimization of a self-amplifying mRNA (saRNA) COVID-19 vaccine, enhanced by co-delivery of B18R-encoding mRNA, it's well-conceived and contains promising data.
Below are some comments:
1. line 46-47, ", saRNA can encode a replicase within cells and use the saRNA as a template to synthesize more RNA copies".how long or how much more copies that saRNA can synthesized compare to conventional non-replicating mRNA? A reference with an average number would be appreciated.
2. line 153 section 2.6 Cells,
a. explain the cell type, something like Huh-7 is hepatocyte-derived carcinoma cell line,
b. the reason that choose Huh-7, pan02, vero cell line as the experiment model.
c. in the following experiments, 293FT cells were used, could you add it to here?
3. line 246, ", with the saRNA3 version demonstrating improved expression "
a. the reason that saRNA3 version demonstrated improved expression?
b. while saRNA3 version may demonstrating improved expression, is there any possible that other version has better outcome in the following evaluation?
4. line 254, ". Consequently, we reduced the transfection dose in following experiments.", A does-dependent experiment is suggested here, optimum performance from 200 ng per well is not guaranteed.
5. figure 2b, Huh7 cell line have results from 48h, 72h, while pan02 have results from 24h, 48h, is this a typo or intended?
6. A fixed dose (2 μg) of B18R-mRNA was used. There’s no dose-response analysis to determine the minimal effective dose or potential toxicity.
7. cellular immunity (CD4+, CD8+ T cell responses), which are critical for interpreting the Th1/Th2 skewing and protective efficacy, is suggested.
8. Figure 4b: IgG titers are higher in mRNA group but NT50 is lower, why mRNA elicit low-affinity antibodies while saRNA could induce high-affinity antibodies?
Reviewer 2 Report
Comments and Suggestions for Authors
This manuscript explored several strategies to improve saRNA vaccine efficacy that include ionizable lipid optimization, removal of impurities from saRNA, and co-delivery of B18R mRNA to suppress type I IFN responses. saRNA represents a novel RNA platform with a promise to elicit more potent RNA translation at a fraction of dose. Yet, the innate immune recognition limits saRNA translation and efficacy. Though the topic is attractive and interesting long-term expression data are presented, there are several key data conflicting with each other. The major and minor concerns are detailed below.
Major concerns:
- There are several conflicting results, which raises concerns about the validity of the experimental system (e.g., quality of saRNA, experimental procedures). Please indicate whether the experiments have been repeated to confirm the conflicting results.
- MFI of EGFP was positively correlated with saRNA dose in Huh7 but negatively correlated with saRNA dose in Pan02.
- ELISA IgG and neutralizing titers showed different trends between mRNA and saRNA.
- 0.5 µg saRNA induced weak neutralizing antibody titer, while 1 µg saRNA induced superior neutralizing antibody titer.
- Western blotting results in Fig.3b/3c/3e are not clear (bands obscure), which affected the evaluation of the work. Better results are needed. Lanes need to be labeled.
- IgG2a/IgG1 ratio more or less than 1 is not a good indicator of Th1 or Th2-biased responses considering absorbance value can be affected by many factors (e.g., serum dilution factors, secondary antibody dilution factors, substrate incubation time). A direct measurement of antigen-specific CD4+ T cells and typical secreting cytokines are better markers for Th1 and Th2 cells.
- There lacks no B18R mRNA control to show B18R mRNA improves saRNA translation or induced immune responses.
Minor concerns:
- Rational to compare the chosen ionizable lipids is needed considering SM102 has been used in commercial vaccine product (Moderna Covid-19 vaccine).
- Cell passage numbers are better mentioned in Fig.2.
- Sample sizes are also needed for Fig.2.
- Does antibody titers on day 7, 14, and 21 show the same trend as Fig.4b/4c on day 42?
- For NT50, it’s better to show absorbance data at different serum dilutions.
Reviewer 3 Report
Comments and Suggestions for Authors
The work of Wang et al. describes several approaches that increase immunogenicity of saRNA-based vaccines. This well-planned comprehensive study can have significant implications for the field. I believe that this manuscript deserves to be published in its present form.
However, the major weakness of this study is the lack of data that support conclusions related to the superior neutralizing activity of antibodies induced by the saRNA vaccine. A protective efficacy study and/or characterization of IgG repertoire would greatly elevate the significance of reported findings.
Round 2
Reviewer 2 Report
Comments and Suggestions for Authors
Thanks for addressing some of my comments. The revised version had no data provided to support incorporation of B18R-encoding mRNA enhanced saRNA transfection, lowered inflammation, or enhanced immune responses as the title stated.
